# Modelling heterogeneous distributions with an Uncountable Mixture of Asymmetric Laplacians

**Axel Brando** [*]
BBVA Data & Analytics
Universitat de Barcelona

**Jose A. Rodríguez-Serrano**[†]
BBVA Data & Analytics

**Jordi Vitrià**[‡]
Universitat de Barcelona

**Alberto Rubio**
BBVA Data & Analytics

## Abstract

In regression tasks, aleatoric uncertainty is commonly addressed by considering a parametric distribution of the output variable, which is based on strong assumptions such as symmetry, unimodality or by supposing a restricted shape. These assumptions are too limited in scenarios where complex shapes, strong skews or multiple modes are present. In this paper, we propose a generic deep learning framework that learns an Uncountable Mixture of Asymmetric Laplacians (UMAL), which will allow us to estimate heterogeneous distributions of the output variable and we show its connections to quantile regression. Despite having a fixed number of parameters, the model can be interpreted as an infinite mixture of components, which yields a flexible approximation for heterogeneous distributions. Apart from synthetic cases, we apply this model to room price forecasting and to predict financial operations in personal bank accounts. We demonstrate that UMAL produces proper distributions, which allows us to extract richer insights and to sharpen decision-making.

## 1 Introduction

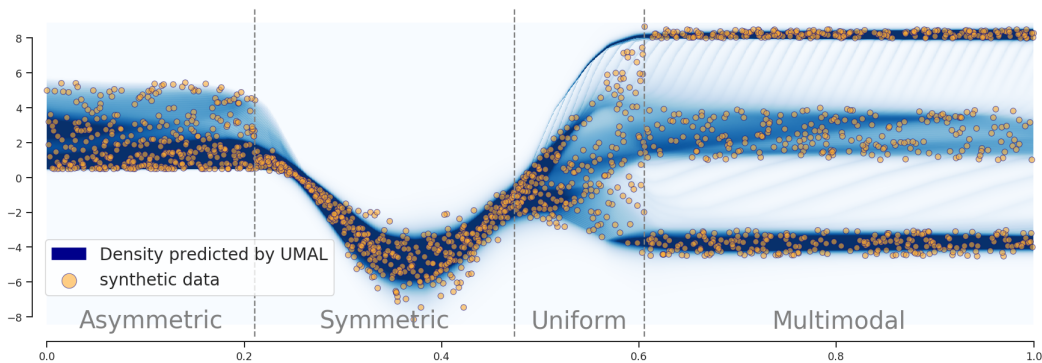

Figure 1: Regression problem with heterogeneous output distributions modelled with UMAL.

In the last decade, deep learning has had significant success in many real-world tasks, such as image classification [1] and natural language processing [2]. While most of the successful examples have been in classification tasks, regression tasks can also be tackled with deep networks by considering architectures where the last layer represents the continuous response variable(s) [3, 4, 5]. However,

[*]axel.brando@bbvadata.com | axelbrando@ub.edu.

[†]joseantonio.rodriguez.serrano@bbvadata.com

[‡]jordi.vitria@ub.edu

this point-wise approach does not provide us with information about the uncertainty underlying the prediction process. When an error in a regression task is associated with a high cost, we might prefer to include uncertainty estimates in our model, or actually estimate the distribution of the response variable.

The modelling of uncertainty in regression tasks has been approached from two main standpoints [6]. On the one hand, one of the sources of uncertainty is "model ignorance", i.e. the mismatch between the model that approximates the task and the true (and unknown) underlying process. This has been referred to as *Epistemic uncertainty*. This type of uncertainty can be modelled using Bayesian methods [7, 8, 9, 10] and can be partially reduced by increasing the size and quality of training data.

On the other hand, another source of uncertainty is "inevitable variability in the response variable", i.e. when the variable to predict exhibits randomness, which is possible even in the extreme case where the true underlying model is known. This randomness could be caused by several factors. For instance, by the fact that the input data do not contain all variables that explain the output. This type of uncertainty has been referred to as *Aleatoric uncertainty*. This can be modelled by considering output distributions [11, 12, 13], instead of point-wise estimations, and is not necessarily reduced by increasing the amount of training data.

We will concentrate on the latter case, our goal being to improve the state-of-the-art in deep learning methods to approximate aleatoric uncertainty. The need to improve current solutions can be understood by considering the regression problem in Figure 1. In this regression problem, the distribution of the response variable exhibits several regimes. Consequently, there is no straightforward definition of aleatoric uncertainty that can represent all these regimes. A quantitative definition of uncertainty valid for one regime (e.g. standard deviation) might not be valid for others. Also, the usefulness of such uncertainty could depend on the end-task. For example, reporting the number of modes would be enough for some applications. For other applications, it might be more interesting to analyse the differences among asymmetries of the predicted distributions.

In this paper, we propose a new model for estimating a *heterogeneous* distribution of the response variable. By *heterogeneous*, we mean that no strong assumptions are made, such as unimodality or symmetry. As Figure 2 shows, this can be done by implementing a deep learning network, $\phi$, which implicitly learns the parameters for the components of an Uncountable

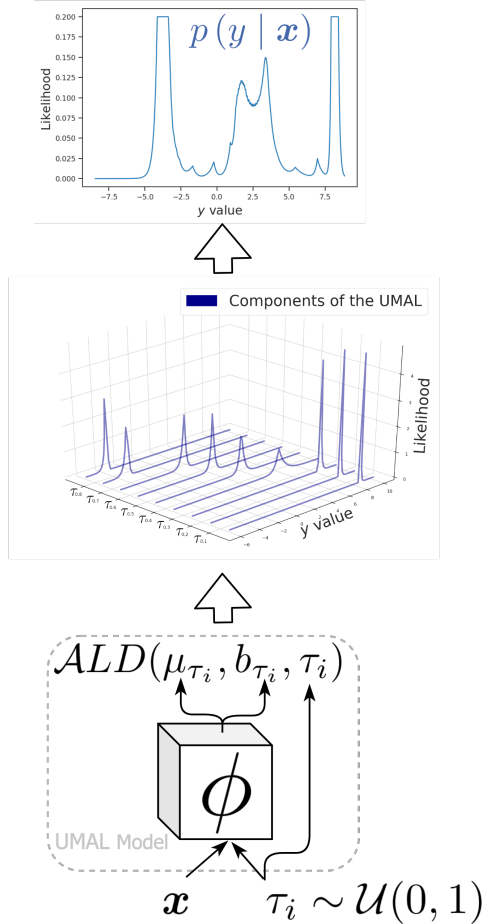

Figure 2: On the bottom we see a representation of the proposed regression model that captures all the components $\tau_i$ of the mixture of Asymmetric Laplacian distributions ($\mathcal{A}LD$) simultaneously. Moreover, this model is agnostic to the architecture of the neural network $\phi$. On the middle, we observe a visualisation of certain $\mathcal{A}LD$ components predicting the upper plot that is the distribution of values of $y$ for a fixed point, $x$, from the *Multimodal* part of Figure 1 (for ease of visualization, the plot has been clipped to $0.2$).

Mixture of Asymmetric Laplacians (UMAL). While the number of weights of such an internal network is finite, we show that it is implicitly fits a mixture of an infinite number of functions.

UMAL is a generic model that is based on two main ideas. Firstly, in order to capture the distribution of possible outputs given the same input, a parametric probability distribution is imposed on the output of the model and a neural network is trained to predict the parameters that maximise the

likelihood of such a probability distribution [11, 14]. Specifically, if that parametric distribution is a mixture model, the approach is known as Mixture Density Network (MDN). And secondly, UMAL can be seen as a generalisation of a method developed in the field of statistics and particularly in the field of econometrics: Quantile Regression (QR) [15]. QR is agnostic with respect to the modelled distribution, which allows it to deal with more heterogeneous distributions. Moreover, QR is still a maximum likelihood estimation of an Asymmetric Laplacian Distribution ($\mathcal{ALD}$) [16]. UMAL extends this model by considering a mixture model that implicitly combines an infinite number of $\mathcal{ALD}$s to approximate the distribution of the response variable.

In order to quantitatively validate the capabilities of the proposed model, we have considered a real problem where the behaviour of the variable to be predicted has a heterogeneous distribution. Furthermore, in the interest of reproducibility we have considered the use of open data [4]. Price forecasting per night of houses / rooms offered on Airbnb, a global online marketplace and hospitality service, fullfils these conditions. Specifically, we predict prices for the cities of Barcelona and Vancouver using public information downloaded from [17]. Price prediction is based on informative features such as neighbourhood, number of beds and other characteristics associated with the houses / rooms. As we can see in the results section, by predicting the full distribution of the price, as opposed to a single estimate of it, we are able to extract much richer conclusions.

Furthermore, we have also applied the UMAL model to a private, large dataset of short sequences, in order to forecast monthly aggregated spending and incomes jointly for each category in personal bank accounts. As in the case of the price prediction per room, to draw conclusions we have used neural networks to perform comparisons using Mixture Density Networks models [11], single distribution estimators as well as other baselines.

## 2 Background concepts and notation

It should be noted that this paper does not attempt to model epistemic uncertainty [6], for which recent work exists related to Bayesian neural networks or its variations [7, 8, 9, 10], by considering a Bayesian interpretation of dropout technique [18] or even combining the forecast of a deep ensemble [19]. Importantly, the main objective of this article is to study models that capture the aleatoric uncertainty in regression problems by using deep learning methods. The reason to focus in this type of uncertainty is because we are interested in problems where there are large volumes of data there but still exists a high variability of possible correct answers given the same input information.

To obtain the richest representation of aleatoric uncertainty, we want to determine a conditional density model $p(y \mid \boldsymbol{x})$ that fits an observed set of samples $\mathcal{D} = (X, Y) = \left\{ (\boldsymbol{x}_i, y_i) \mid \boldsymbol{x}_i \in \mathbb{R}^F, y_i \in \mathbb{R} \right\}_{i=1}^{n}$, which we assumed to be sampled from an unknown distribution $q(y \mid \boldsymbol{x})$. To achieve this goal, we consider the solutions that restrict $p$ to a parametric family distributions $\{ f(y; \boldsymbol{\theta}) \mid \boldsymbol{\theta} \in \Theta \}$, where $\boldsymbol{\theta}$ denotes the parameters of the distributions [20]. These parameters are the outputs of a deep learning function $\phi \colon \mathbb{R}^F \to \Theta$ with weights $\boldsymbol{w}$ optimised to maximise the likelihood function in a training subset of $\mathcal{D}$. The problem appears when the assumed parametric distribution imposed on $p$ differs greatly from the real distribution shape of $q$. This case will become critical the more *heterogeneous* $q$ is with respect to $p$, i.e. in cases when its distribution shape is more complex, containing further behaviours such as extra modes or stronger asymmetries.

## 3 Modelling heterogeneous distributions

In this paper, we have selected as baseline approaches two different types of distribution that belong to the generalised normal distribution family: the normal and the Laplacian distribution. Thus, in both cases the neural network function is defined as $\phi \colon \mathbb{R}^F \to \mathbb{R} \times (0, +\infty)$ to predict location and scale parameters. However, the assumption of a simple normal or Laplace variability in the output of the model forces the conditional distribution of the output given the input to be unimodal [11]. This could be very limiting in some problems, such as when we want to estimate the price of housing and there may be various types of price distributions given the same input characteristics.

**Mixture Density Network**   One proposed solution in the literature to approximate a multimodal conditional distribution is the Mixture of Density Networks (MDN) [11]. Specifically, the mixture likelihood for any normal or Laplacian distribution components is

$$p\left(y \mid x, \boldsymbol{w}\right) = \sum_{i=1}^{m} \alpha_i(x) \cdot pdf\left(y \mid \mu_i(x), b_i(x)\right), \tag{1}$$

where $m$ denotes the fixed number of components of the mixture, each one being defined by the distribution function $pdf$. On the other hand, $\alpha_i(x)$ would be the mixture weight (such that $\sum_i^m \alpha_i(x) = 1$). Therefore, for this type of model we will have an extra output to predict, $\alpha$, in the original neural network, i.e. $\phi\colon \mathbb{R}^F \to \mathbb{R}^m \times (0, +\infty)^m \times [0,1]^m$.

Although this model can approximate any type of distribution, provided $m$ is sufficiently large [21], it does not capture asymmetries at the level of each component. Additionally, it entails determining the optimal number of components $m$ e.g. by cross-validation [22], which in practice multiplies the training cost by a significant factor.

**Quantile Regression**   Alternatively, an extension to classic regression has been proposed in the field of statistics and econometrics: Quantile Regression (QR). QR is based on estimating the desired conditional quantiles of the response variable [15, 23, 24, 25]. Given $\tau \in (0, 1)$, the $\tau$-th quantile regression loss function would be defined as

$$\mathcal{L}_\tau(x, y; \boldsymbol{w}) = (y - \mu(x)) \cdot (\tau - \mathbb{1}[y < \mu(x)]), \tag{2}$$

where $\mathbb{1}[p]$ is the indicator function that verifies the condition $p$. This loss function is an asymmetric convex loss function that penalises overestimation errors with weight $\tau$ and underestimation errors with weight $1 - \tau$ [26].

Recently, some works have combined neural networks with QR [26, 12, 27]. For instance, in the reinforcement learning field, a neural network has been proposed to approximate a given set of quantiles [26]. This is achieved by jointly minimizing a sum of terms like those in Equation 2, one for each given quantile. Following this, the Implicit Quantile Networks (IQN) model was proposed [12] in order to learn the full quantile range instead of a finite set of quantiles. This was done by considering the $\tau$ parameter as an input of the deep reinforcement learning model and conditioning the single output to the input desired quantile, $\tau$. In order to optimize for all possible $\tau$ values, the loss function considers an expectation over $\tau$, which in the stochastic gradient descent method is approximated by sampling $\tau \sim \mathcal{U}(0, 1)$ from a uniform distribution in each iteration. Recently, a neural network has also been applied to regression problems in order to simultaneously minimise the Equation 2 for all quantile values sampled as IQN [13]. Thus, both solutions consider a joint but "independent" quantile minimization with respect to the loss function. Consequently, for the sake of consistency with the following nomenclature, we will refer to them as *Independent QR* models.

Given a neural network function, $\phi\colon \mathbb{R}^{F+1} \to \mathbb{R}$, where the input is $(\boldsymbol{x}, \tau) \in \mathbb{R}^{F+1}$, such that implicitly approximates all the quantiles $\tau \in (0, 1)$, we can obtain the distribution shape for a given input $\boldsymbol{x}$ by integrating the conditioned function over $\tau$. However, due to the fact that this function is estimated empirically, there is no guarantee that it will be strictly increasing with respect to the value $\tau$ and this can lead to a *crossing quantiles* phenomenon [23, 13]. Below, we introduce a concept that allows this limitation to be bypassed by applying a method described in the following section.

**Asymmetric Laplacian distribution**   As it is widely known, when a function is fitted using the mean square, or the mean absolute error loss, it is equivalent to a maximum likelihood estimation of the location parameter of a Normal distribution, or Laplacian distribution, respectively. Similar to these unimodal cases, when we minimise Equation 2, we are optimising the maximum likelihood of the location parameter of an Asymmetric Laplacian Distribution ($\mathcal{A}$LD) [16, 23] expressed as

$$\mathcal{A}LD\left(y \mid \mu, b, \tau\right) = \frac{\tau(1-\tau)}{b(x)} \exp\left\{-\left(y - \mu(x)\right) \cdot \left(\tau - \mathbb{1}[y < \mu(x)]\right)/b(x)\right\}. \tag{3}$$

When $\mu, b$ parameters are predicted by using deep networks conditioned to $\tau$, we are considering a non-point-wise approach of QR. Next, we combine all $\mathcal{A}LD$s to infer a response variable distribution.

# 4 The Uncountable Mixture of Asymmetric Laplacians model

In order to define the proposed framework, the objective is to consider a model that corresponds to the mixture distribution of all possible $\mathcal{A}LD$ functions with respect to the asymmetry parameter, $\tau \in (0,1)$. This mixture model has an uncountable set of components that are combined to produce the uncountable mixture[5] distribution. Let $\boldsymbol{w}$ be the weights of the deep learning model to estimate, $\phi \colon \mathbb{R}^{F+1} \to \mathbb{R} \times (0, +\infty)$, which predicts the $(\mu_\tau, b_\tau)$ parameters of the different $\mathcal{A}LD$s conditioned to a $\tau$ value. Then, we can consider the following compound model marginalising over $\tau$:

$$p\left(y \mid x, \boldsymbol{w}\right) = \int \alpha_\tau(x) \cdot \mathcal{A}LD\left(y \mid \mu_\tau(x), b_\tau(x), \tau\right) \, d\tau. \tag{4}$$

Now we can make two considerations. On the one hand, we assume a uniform distribution for each component $\alpha_\tau$ of the mixture model. Therefore, the weight $\alpha_\tau$ is the same for all $\mathcal{A}LD$s, maintaining the restriction to integrate to $1$. On the other hand, in order to make the integral tractable at the time of training, following the strategy proposed in implicit cases [12, 13], we consider a random variable $\tau \sim \mathcal{U}(0,1)$ and apply Monte Carlo (MC) integration [7], selecting $N_\tau$ random values of $\tau$ in each iteration, so that we discretise the integral. This results in the following expression:

$$p\left(y \mid x, \boldsymbol{w}\right) \approx \frac{1}{N_\tau} \sum_{t=1}^{N_\tau} \mathcal{A}LD(y \mid \mu_{\tau_t}(x), b_{\tau_t}(x), \tau_t). \tag{5}$$

Therefore, the Uncountable Mixture of Asymmetric Laplacians (UMAL) model is optimised by minimising the following negative log-likelihood function with respect to $\boldsymbol{w}$,

$$-\log p\left(Y \mid X, \boldsymbol{w}\right) \approx -\sum_{i=1}^{n} \log \left( \sum_{t=1}^{N_\tau} \exp \left[\log \mathcal{A}LD(y_i \mid \mu_{\tau_t}(x_i), b_{\tau_t}(x_i), \tau_t)\right] \right) - \log(N_\tau), \tag{6}$$

where, as is commonly considered in mixture models [29], we have a "logarithm of the sum of exponentials". This form allows application of the LogSumExp Trick [30] during optimisation to prevent overflow or underflow when computing the logarithm of the sum of the exponentials.

## 4.1 Connection with quantile models

It is important to note the link between UMAL and QR. If we consider an *Independent QR* model where the entire range of quantiles is implicitly and independently approximated (as in the case of IQN), then the mode of an $\mathcal{A}LD$ can be directly inferred. Thus, in inference time there is a perfect solution that estimates the real distribution but in a point-estimate manner. However, an alternative approach would be to minimise all the negative logarithm of $\mathcal{A}LD$s as a sum of distributions where each one "independently" captures the variability for each quantile. This solution is, in fact, an upper bound of the UMAL model. Applying Jensen's Inequality to the negative logarithm function of Equation 6 gives us an expression that corresponds to consider all $\mathcal{A}LD$s as independent elements,

$$-\log p\left(Y \mid X, \boldsymbol{w}\right) \leq -\sum_{i=1}^{n} \left( \sum_{t=1}^{N_\tau} \log \mathcal{A}LD(y_i \mid \mu_{\tau_t}(x_i), b_{\tau_t}(x_i), \tau_t) \right) - \log(N_\tau). \tag{7}$$

We will refer to this upper bound solution as *Independent $\mathcal{A}LD$* and it will be used as a baseline in further comparisons.

# 5 UMAL as a deep learning framework

UMAL can be viewed as a framework for upgrading any point-wise estimation regression model in deep learning to an output distribution shape forecaster, as show in Algorithm 2. This implementation can be performed using any automatic differentiation library such as TensorFlow [31] or PyTorch [32]. Additionally, it also performs the Monte Carlo step within the procedure, which results in more efficient computation in training time.

Therefore, in order to obtain the conditioned mixture distribution we should perform Algorithm 3. By using this rich information we are able to conduct the following experiments.

---

**Prerequisites 1** Definitions and functions used for following Algorithms

▷ $x$ has batch size and number of features as shape, $[bs, F]$.
▷ RESHAPE( $tensor, shape$): returns $tensor$ with shape $shape$.
▷ REPEAT($tensor, n$): repeats last dimension of $tensor$ $n$ times.
▷ CONCAT($T_1, T_2$): concat $T_1$ and $T_2$ by using their last dimension.
▷ LEN($T_1$): number of elements in $T_1$.

---

**Algorithm 2** How to build UMAL model by using any deep learning architecture for regression

1: **procedure** BUILD_UMAL_GRAPH(input vectors $x$, deep architecture $\phi$, MC sampling $N_\tau$)
2:      $x \leftarrow$ RESHAPE( REPEAT($x, N_\tau$), $[bs \cdot N_\tau, F]$)    ▷ Adapting $x$ to be able to associate a $\tau$.
3:      $\tau \leftarrow \mathcal{U}(0, 1)$                                  ▷ $\tau$ must have $[bs \cdot N_\tau, 1]$ shape.
4:      $i \leftarrow$ CONCAT($x, \tau$)                      ▷ The $i$ has $[bs \cdot N_\tau, F + 1]$ shape.
5:      $(\mu, b) \leftarrow \phi_\tau(i)$                       ▷ Applying any deep learning function $\phi$.
6:      $\mathcal{L} \leftarrow$ Equation 6         ▷ Applying the UMAL Loss function by using the $(\mu, b, \tau)$ triplet.
7:      **return** $\mathcal{L}$

---

**Algorithm 3** How to generate the final conditioned distribution by using UMAL model

1: **procedure** PREDICT(input vectors $x$, response vectors $y$, deep architecture $\phi$, selected $\tau$s $sel_\tau$)
2:      $\tau \leftarrow$ RESHAPE( REPEAT($sel_\tau, bs$), $[bs \cdot \text{LEN}(sel_\tau), 1]$)      ▷ Adapting $\tau$ shape.
3:      $x \leftarrow$ RESHAPE( REPEAT($x, sel_\tau$), $[bs \cdot \text{LEN}(sel_\tau), F]$)      ▷ Adjusting $x$ shape.
4:      $i \leftarrow$ CONCAT($x, \tau$)                    ▷ The $i$ has $[bs \cdot N_\tau, F + 1]$ shape.
5:      $(\mu, b) \leftarrow \phi_\tau(i)$                 ▷ Apply the trained deep learning function $\phi$.
6:      $p(y \mid x) \leftarrow \frac{1}{N_\tau} \sum_{t=1}^{N_\tau} \mathcal{ALD}(y \mid \mu_{\tau_t}, b_{\tau_t}, \tau_t)$      ▷ Calculate mixture model of $sel_\tau$ for each $y$.
7:      **return** $p(y \mid x)$

---

# 6 Experimental Results

## 6.1 Data sets and experiment settings

In this section, we show the performance of the proposed model. All experiments are implemented in TensorFlow [33] and Keras [34], running in a workstation with Titan X (Pascal) GPU and GeForce RTX 2080 GPU. Regarding parameters, we use a common learning rate of $10^{-3}$. In addition, to restrict the value of the scale parameter, $b$, to strictly positive values, the respective output have a softplus function [35] as activation. We will refer to the number of parameters to be estimated as $P$. On the other hand, the Monte Carlo sampling number, $N_\tau$, for Independent QR, $\mathcal{ALD}$ and UMAL models will always be fixed to 100 at the training time. Furthermore, all public experiments are trained using an early stopping training policy with 200 epochs of patience for all compared methods.

**Synthetic regression** Figure 1 corresponds to the following data set. Given $(X, Y) = \{(x_i, y_i)\}_{i=1}^{3800}$ points where $x_i \in [0, 1]$ and $y_i \in \mathbb{R}$, they are defined by 4 different fixed synthetic distributions depending on the $X$ range of values. In particular, if $x_i < 0.21$, then the corresponding $y_i$ came from a Beta($\alpha = 0.5, \beta = 1$) distribution. Next, if $0.21 < x_i < 0.47$, then their $y_i$ values

Table 1: Comparison of the Log-Likelihood of the test set over different alternatives to model the distribution of the different proposed data sets. The scale for each data set is indicated in parenthesis.

**Log-Likelihood comparison**

| Model | Synthetic $(10^2)$ | BCN RPF $(10^3)$ | YVC RPF $(10^2)$ | Financial $(10^6)$ |
|---|---|---|---|---|
| Normal distribution | $-39.88 \pm 13.4$ | $-38.44 \pm 6.55$ | $-70.79 \pm 3.26$ | $-8.56$ |
| Laplace distribution | $-41.30 \pm 0.78$ | $-19.84 \pm 0.93$ | $-82.87 \pm 8.01$ | $-7.88$ |
| Independent QR | $-119.0 \pm 7.68$ | $-32.98 \pm 1.63$ | $-113.54 \pm 10.4$ | $-8.26$ |
| 2 comp. Normal MDN | $-43.14 \pm 6.12$ | $-28.59 \pm 3.38$ | $-74.11 \pm 3.26$ | $-6.37$ |
| 3 comp. Normal MDN | $-51.79 \pm 21.0$ | $-31.66 \pm 4.85$ | $-74.22 \pm 2.37$ | $-7.25$ |
| 4 comp. Normal MDN | $-111.6 \pm 43.27$ | $-28.60 \pm 7.22$ | $-76.85 \pm 5.95$ | $-6.75$ |
| 10 comp. Normal MDN | $-184.3 \pm 35.5$ | $-27.72 \pm 2.81$ | $-77.26 \pm 6.12$ | $-10.40$ |
| 2 comp. Laplace MDN | $-42.83 \pm 1.54$ | $-19.76 \pm 0.18$ | $-65.52 \pm 0.40$ | $-10.83$ |
| 3 comp. Laplace MDN | $-64.13 \pm 36.70$ | $-19.57 \pm 0.30$ | $-78.80 \pm 3.79$ | $-5.84$ |
| 4 comp. Laplace MDN | $-52.53 \pm 8.79$ | $-19.89 \pm 0.44$ | $-66.58 \pm 1.10$ | $-5.72$ |
| 10 comp. Laplace MDN | $-155.9 \pm 32.9$ | $-21.45 \pm 0.83$ | $-82.51 \pm 9.66$ | $-6.28$ |
| Independent $\mathcal{ALD}$ | $-39.03 \pm 0.45$ | $-19.03 \pm 0.81$ | $-64.16 \pm 0.19$ | $-5.66$ |
| **UMAL model** | $\mathbf{-28.14 \pm 0.44}$ | $\mathbf{-18.04 \pm 0.72}$ | $\mathbf{-62.68 \pm 0.21}$ | $\mathbf{-5.49}$ |

are obtained from $\mathcal{N}(\mu = 3 \cdot \cos x_i - 2, \sigma =\mid 3 \cdot \cos x_i - 2 \mid)$ distribution depending on $x_i$ value. Then, when $0.47 < x_i < 0.61$ their respective $y_i$ values is obtained from an increasing uniform random distribution and, finally, all values above $0.61$ are obtained from three different uniform distributions: $\mathcal{U}(8, 0.5)$, $\mathcal{U}(1, 3)$ and $\mathcal{U}(-4.5, 1.5)$. A total of $50\%$ of the random uniform generated data were considered as test data, $40\%$ for training and $10\%$ for validation.

For all compared models, we will use the same neural network architecture for $\phi$. This consists of 4 dense layers that have output dimensions 120, 60, 10 and $P$, respectively, and all but the last layer with ReLu activation. Regarding training time, all models took less than 3 minutes to converge.

**Room price forecasting (RPF)** By using the publicly available information from the the Inside Airbnb platform [17] we selected Barcelona (BCN) and Vancouver (YVC) as the cities to carry out the comparison of the models in a real situation. For both cities, we select the last time each house or flat appeared within the months available from April 2018 to March 2019.

The regression problem will be defined as predicting the real price per night of each flat in their respective currency using the following information: the one hot encoding of the categorical attributes (present in the corresponding Inside Airbnb "listings.csv" files) of district number, neighbourhood number, room type and property type, as well as the number of bathrooms, accommodates values together with the latitude and longitude normalised according to the minimums and maximums of the corresponding city.

Given the $36,367$ and $11,497$ flats in BCN and YVC respectively, we have considered $80\%$ as a training set, $10\%$ as a validation set and the remaining $10\%$ as a test set. Regarding the trained models, all share the same neural network architecture for their $\phi$, composed of 6 dense layers with ReLu activation in all but the last layer and their output dimensions of 120, 120, 60, 60, 10 and $P$, respectively. Concerning training time, all models took less than 30 minutes to converge.

**Financial estimation** The aim here is to anticipate personal expenses and income for each specific financial category in the upcoming month by only considering the last 24 months of aggregated historic values for that customer as a short-time series problem. This private data set contains monthly aggregated expense and income operations for each costumer in a certain category as time series of 24 months. 1.8 millions of that time series of a selected year will be the training set, 200 thousand will be the validation set and 1 million time series of the following year will be the test set. Regarding the $\phi$ architecture for all compared models, after an internal previous refinement task to select the best architecture, we used a recurrent model that contains 2 concatenated Long Short-Term Memory (LSTM) layers [36] of 128 output neurons each, and then two dense layers of 128 and $P$ outputs, respectively. It is important to note that because all compared solutions used for this article are agnostic with respect to the architecture, the only decision we need to take is how to insert the extra $\tau$

Figure 3: Plot with the performance of three different models in terms of calibration. The mean and standard deviation for all folds of the mean absolute error between the predicted calibration and the perfect ideal calibration is represented in the table.

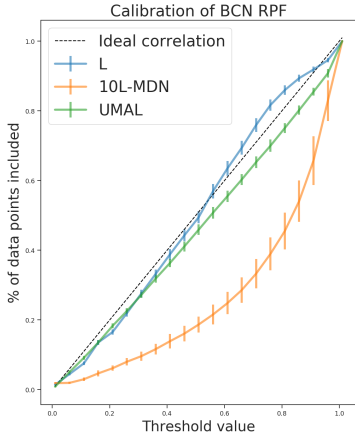

| **Likelihood calibration comparison** | | |
|---|---|---|
| Model | BCN RPF | YVC RPF |
| Normal distribution | $.12 \pm .04$ | $\mathbf{.04 \pm .01}$ |
| Laplace distribution | $\mathbf{.03 \pm .00}$ | $.06 \pm .01$ |
| Independent QR | $.10 \pm .02$ | $.12 \pm .02$ |
| 2 comp. Normal MDN | $\mathbf{.05 \pm .02}$ | $.12 \pm .05$ |
| 3 comp. Normal MDN | $.07 \pm .02$ | $.14 \pm .04$ |
| 4 comp. Normal MDN | $.10 \pm .03$ | $.17 \pm .06$ |
| 10 comp. Normal MDN | $.19 \pm .04$ | $.19 \pm .06$ |
| 2 comp. Laplace MDN | $.05 \pm .01$ | $.09 \pm .01$ |
| 3 comp. Laplace MDN | $.08 \pm .02$ | $.11 \pm .02$ |
| 4 comp. Laplace MDN | $.13 \pm .05$ | $.12 \pm .03$ |
| 10 comp. Laplace MDN | $.24 \pm .03$ | $.18 \pm .05$ |
| Independent $\mathcal{ALD}$ | $.06 \pm .01$ | $\mathbf{.02 \pm .01}$ |
| UMAL model | $\mathbf{.04 \pm .01}$ | $.07 \pm .01$ |

information into the $\phi$ function in the QR, $\mathcal{ALD}$ and UMAL models. In these cases, for simplicity, we add the information $\tau$ repeatedly as one more attribute of each point of the input time series.

## 6.2 Results

**Log-Likelihood comparison** We compared the log-likelihood adaptation of all models presented in Table 1 for the three type of problems introduced. For all public data sets, we give their corresponding mean and standard deviation over the 10 runs of each model we did. Due to computational resources, the private data set is the result after one execution per model. Furthermore, we take into account different numbers of components for the different MDN models. We observe that the best solutions for MDN are far from the UMAL cases. Thus, we conclude that the UMAL models achieve the best performance in all of these heterogeneous problems.

**Calibrated estimated likelihoods** To determine whether the learned likelihood is useful (i.e. if UMAL yields calibrated outputs), we performed an additional empirical study to assess this point. We highlight that our system predicts an output distribution $p(y|x, \boldsymbol{w})$ (not a confidence value). Specifically, we have computed the % of actual test data that falls into different thresholds of predicted probability. Ideally, given a certain threshold $\theta \in [0, 1]$, the amount of data points with a predicted probability above or equal to $1 - \theta$ should be similar to $\theta$. On the left side of Figure 3 we plot these measures for different methods (in green, our model) when considering the BCN RPF dataset. Furthermore, on the right side of Figure 3 we report the mean absolute error between the empirical measures and the ideal ones for both rental-price data sets. As we can see, the conditional distribution predicted by UMAL has low error values. Therefore, we can state that UMAL produces proper and calibrated conditional distributions that are especially suitable for heterogeneous problems[6].

**Predicted distribution shape analysis** From right to left in Figure 4 we show a 50 perplexity with Wasserstein distance t-SNE [37] projection from 500 linearly spaced discretisation of the normalised predicted distribution to 2 dimensions for each room of the test set in Barcelona. Each colour of the palette corresponds to a certain DBSCAN [38] cluster obtained with $\epsilon = 5.8$ and 40 minimum samples as DBSCAN parameters. We show a Hex-bin plot over the map of Barcelona, where the colours correspond to the mode cluster of all the rooms inside the hexagonal limits. A similar study would be useful to extract patterns inside the city, and consequently adapt specific actions to them.

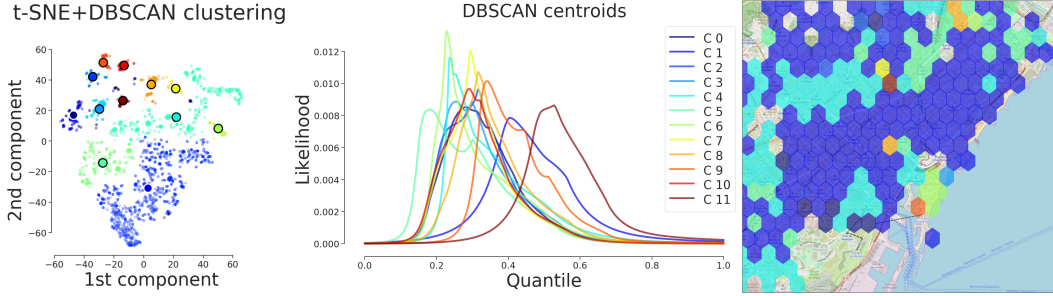

Figure 4: DBSCAN clustering of the t-SNE projection to 2 dimensions of normalised Barcelona predicted distributions. Hexbin plot of most common cluster for each hexagon on top of the map.

# 7 Conclusion

This paper has introduced the Uncountable Mixture of Asymmetric Laplacians (UMAL) model, a framework that uses deep learning to estimate output distribution without strong restrictions (Figure 1). As shown in Figure 2, UMAL is a model that implicitly learns infinite $\mathcal{ALD}$ distributions, which are combined to form the final mixture distribution. Thus, in contrast with mixture density networks, UMAL does not need to increase its internal neural network output, which tends to produce unstable behaviours when it is required. Furthermore, the Monte Carlo sampling of UMAL could be considered as a batch size that can be updated even during training time.

We have presented a benchmark comparison in terms of log-likelihood adaptation in the test set of three different types of problems. The first was a synthetic experiment with distinct controlled heterogeneous distributions that contains multimodality and skewed behaviours. Next, we used public data to create a complex problem for predicting the room price per night in two different cities as two independent problems. Finally, we compared all the presented models in a financial forecasting problem anticipating the next monetary aggregated monthly expense or income of a certain customer given their historical data. We showed that the UMAL model outperforms the capacity to approximate the output distribution with respect to the other baselines as well as yielding calibrated outputs.

In introducing UMAL we emphasise the importance of taking the concept of aleatoric uncertainty to a whole richer level, where we are not restricted to only studying variability or evaluating confidence intervals to make certain actions but can carry out shape analysis in order to develop task-tailored methods.

**Acknowledgements**   We gratefully acknowledge the Government of Catalonia's Industrial Doctorates Plan for funding part of this research. The UB acknowledges that part of the research described in this chapter was partially funded by RTI2018-095232-B-C21 and SGR 1219. We would also like to thank BBVA Data and Analytics for sponsoring the industrial PhD.

## Footnotes

[4]The source code to reproduce the public results reported is published in https://github.com/BBVA/UMAL.

[5]The concept of "uncountable mixture" refers to the marginalisation formula that defines a compound probability distribution [28].

[6]In the Appendix section, we have evaluated calibration quality and negative log-likelihood on the UCI data sets with the same architectures as [19, 8].

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
