[Supplementary Material]

# A   Additional results

In this section, we propose an empirical study of uncertainty quality over standard regression benchmarks such as the UCI datasets presented in [8, 18, 19]. Similarly to Figure 3, in Table 2 we have reported the mean absolute error between the amount of data points above or equal to a certain predicted probability and the ideal one. As shown here, UMAL still produces proper and calibrated conditional distribution.

Table 2: Mean and standard deviation for all folds of the mean absolute error between the predicted calibration and the perfect ideal calibration.

| | Housing | Concrete | Energy | Kin8nm | Naval | Power | Protein | Wine | Yacht |
|---|---|---|---|---|---|---|---|---|---|
| N | **.08 ± .04** | **.03 ± .01** | **.03 ± .01** | **.02 ± .01** | .39 ± .02 | **.02 ± .01** | .03 ± .00 | **.03 ± .01** | **.06 ± .02** |
| L | **.07 ± .04** | .05 ± .02 | **.04 ± .01** | .04 ± .01 | .35 ± .03 | .05 ± .01 | **.02 ± .00** | **.06 ± .02** | **.07 ± .02** |
| I-QR | .20 ± .05 | .18 ± .02 | .15 ± .04 | .17 ± .01 | **.12 ± .05** | .20 ± .02 | .06 ± .01 | .19 ± .03 | .14 ± .06 |
| 2N-MDN | **.07 ± .05** | **.04 ± .02** | .05 ± .02 | **.01 ± .01** | .36 ± .04 | **.03 ± .01** | .06 ± .01 | .08 ± .03 | .06 ± .02 |
| 3N-MDN | **.07 ± .05** | **.07 ± .03** | .04 ± .02 | **.02 ± .01** | .37 ± .04 | **.03 ± .01** | .11 ± .01 | .15 ± .04 | **.07 ± .02** |
| 4N-MDN | .09 ± .05 | .10 ± .03 | **.05 ± .02** | **.03 ± .01** | .36 ± .05 | **.03 ± .01** | .15 ± .01 | .18 ± .03 | **.07 ± .04** |
| 10N-MDN | .12 ± .06 | .22 ± .06 | .09 ± .04 | .08 ± .01 | .33 ± .05 | **.03 ± .01** | .22 ± .01 | .18 ± .02 | **.09 ± .05** |
| 2L-MDN | .09 ± .05 | **.06 ± .02** | **.05 ± .02** | .04 ± .01 | .32 ± .04 | .07 ± .02 | .07 ± .00 | **.06 ± .02** | **.06 ± .03** |
| 3L-MDN | **.11 ± .05** | .10 ± .03 | .08 ± .03 | .05 ± .01 | .29 ± .04 | .08 ± .02 | .12 ± .01 | .16 ± .03 | **.06 ± .02** |
| 4L-MDN | **.14 ± .06** | .12 ± .03 | **.08 ± .04** | .06 ± .01 | .31 ± .04 | .07 ± .02 | .15 ± .01 | .15 ± .02 | **.05 ± .02** |
| 10L-MDN | .21 ± .05 | .18 ± .04 | .16 ± .05 | .11 ± .01 | .27 ± .06 | .08 ± .02 | .22 ± .01 | .17 ± .01 | **.10 ± .04** |
| I-$\mathcal{ALD}$ | **.07 ± .06** | **.04 ± .01** | **.05 ± .02** | .04 ± .01 | .44 ± .01 | **.04 ± .01** | .07 ± .00 | **.03 ± .01** | .09 ± .04 |
| UMAL | .10 ± .05 | .07 ± .04 | **.06 ± .02** | **.02 ± .01** | .43 ± .01 | **.02 ± .01** | **.02 ± .00** | .13 ± .06 | **.06 ± .03** |

For the sake of completeness, we have also computed the UMAL negative log-likelihood for UCI datasets (see Table 3) following [19]. These results re-emphasise that UMAL is in the best positions. However, it should be noted that most of these databases have a **small sample size** and that aleatoric uncertainty cannot be reliably estimated in this regime. We hypothesize that a better solution would be to simultaneously estimate epistemic (as in [18, 19, 8]) and aleatoric uncertainty.

Table 3: Comparison of the Negative Mean Log-Likelihood of the test set over different train-test folds proposed in [8].

| | Housing | Concrete | Energy | Kin8nm | Naval | Power | Protein | Wine | Yacht |
|---|---|---|---|---|---|---|---|---|---|
| N | 2.76 ± .34 | **3.20 ± .16** | **2.13 ± .24** | −1.15 ± .03 | −3.67 ± .01 | **2.83 ± .03** | 2.84 ± .03 | 1.05 ± .14 | **1.86 ± .31** |
| L | **2.59 ± .20** | **3.21 ± .13** | **2.06 ± .20** | −1.08 ± .04 | −3.73 ± .04 | **2.87 ± .03** | 2.74 ± .01 | 1.00 ± .08 | **1.54 ± .37** |
| I-QR | 10.96 ± 2.4 | 10.19 ± .95 | 9.45 ± 1.3 | 9.22 ± .66 | 5.14 ± .89 | 8.39 ± .45 | 8.14 ± .52 | 12.30 ± .91 | 10.32 ± 2.9 |
| 2N-MDN | 2.74 ± .30 | 3.25 ± .21 | **2.02 ± .30** | −1.15 ± .05 | −3.66 ± .02 | **2.85 ± .05** | 2.56 ± .03 | 1.33 ± .61 | **1.55 ± .32** |
| 3N-MDN | 2.68 ± .28 | 3.64 ± .28 | **2.30 ± .43** | −1.15 ± .05 | −3.66 ± .01 | **2.85 ± .04** | 2.90 ± .15 | 0.69 ± 1.0 | **1.54 ± .52** |
| 4N-MDN | 2.87 ± .46 | 3.74 ± .28 | 2.46 ± .39 | **−1.12 ± .04** | **−3.66 ± .03** | **2.86 ± .05** | 3.32 ± .11 | **0.52 ± .90** | **1.43 ± .36** |
| 10N-MDN | 3.10 ± .46 | 5.64 ± 1.1 | 3.03 ± .71 | −0.99 ± .06 | −3.64 ± .03 | **2.86 ± .04** | 4.94 ± .75 | 0.75 ± .95 | **1.75 ± .49** |
| 2L-MDN | 2.61 ± .23 | **3.28 ± .14** | **2.06 ± .30** | −1.10 ± .04 | −3.70 ± .06 | 2.91 ± .05 | 2.50 ± .03 | 0.59 ± .63 | **1.37 ± .42** |
| 3L-MDN | 2.65 ± .25 | 3.45 ± .16 | **2.30 ± .21** | −1.09 ± .03 | **−3.66 ± .06** | 2.95 ± .04 | 2.65 ± .06 | **−0.81 ± .70** | **1.39 ± .35** |
| 4L-MDN | 2.76 ± .42 | 3.57 ± .14 | **2.31 ± .35** | −1.10 ± .05 | **−3.68 ± .06** | 2.93 ± .04 | 2.79 ± .08 | **−0.65 ± .96** | **1.45 ± .35** |
| 10L-MDN | 3.17 ± .46 | 3.95 ± .34 | 2.80 ± .49 | −0.98 ± .07 | −3.62 ± .10 | 2.96 ± .05 | 3.46 ± .12 | 0.52 ± .74 | **1.63 ± .34** |
| I-$\mathcal{ALD}$ | 2.79 ± .56 | 3.87 ± .12 | **2.28 ± .11** | −1.00 ± .05 | −2.82 ± .01 | **2.89 ± .02** | 2.68 ± .01 | 1.01 ± .07 | **1.78 ± .41** |
| UMAL | **2.59 ± .26** | 3.74 ± .15 | **2.13 ± .14** | **−1.09 ± .03** | −2.81 ± .01 | **2.85 ± .03** | **2.40 ± .01** | **0.14 ± .70** | **1.41 ± .38** |