[Reviews · NeurIPS 2019]

Reviewer 1



## Update ## I have read the rebuttal and would like to thank the authors for the new experimental results they have included. The additional results are very helpful for evaluating the method, although I would have liked to see the a similar plot as Figure 3 in Tagasovska and Lopez-Paz [1]. I find the calibration of UMAL predictions on room-price forecasting for BCN quite convincing. These results, along with the calibration on UCI, have resolved my concerns about calibration of the UMAL method. The test log-likelihoods on UCI are less interesting, but it is good that UMAL performs as expected. For instance, the fact that UMAL outperforms the similar Independent ALD method is nice to see. Overall, I think that the additional experiments make a good argument for accepting the submission. This said, I believe that my initial review may have been overly positive and so, given new results, I have decided to maintain my score of 7. ## Initial Review ## The authors examine uncertainty modeling in regression problems with potentially complex target distributions. They propose learning asymmetric Laplace distributions over targets in order to better capture the uncertainty stemming from the data-generating process. The scale and location parameters of the asymmetric Laplace distributions are parameterized by a neural network, which also takes the asymmetry parameter as an input. The authors treat the asymmetry parameter as a latent variable with a uniform distribution, which, by integration, yields a "uncountable" mixture of asymmetric Laplace distributions. The parameterizing network is learned by maximizing the marginal-likelihood, which is approximated using Monte Carlo integration. Related work on simultaneous quantile regression [1] with neural networks is shown to maximize a lower bound on the proposed marginal-likelihood objective with respect to the location parameter [1, 2]. The proposed uncountable mixture of asymmetric Laplacians (UMAL) model is compared to finite-mixtures models and methods for quantile regression on a synthetic dataset, two rental-price datasets, and a financial time-series problem. The proposed method outperforms these approaches with respect to test log-likelihood. Originality: The paper is an incremental advance on recent work for quantile regression with neural networks [1,2]. The major differences from simultaneous quantile regression [1] are (1) the scale parameter of the asymmetric Laplacian is also parameterized; and (2) the asymmetry parameter is (approximately) marginalized out. In comparison, simultaneous quantile regression trains the parameterizing network in expectation over the scale parameter using SGD. These changes appear to have a meaningful impact on experimental performance. Clarity: The writing is clear and well-organized and I enjoyed reading the submission. However, there are several typos whose correction will improve the flow of the paper. See minor comments below. Quality and Significance: This paper synthesizes recent work on quantile regression and provides a clear, probabilistic interpretation for these approaches. This discussion is useful for the community. The progression from simultaneous quantile regression to UMAL was straightforward, but is a good contribution in my opinion. It might show the effectiveness of the proposed method better if additional experimental results were presented. Further comparisons could be done on regression datasets from the UCI repository [3]. An empirical study of uncertainty quality would also make the experimental section much more convincing. For instance, Tagasovska and Lopez-Paz [1] evaluate the quality of uncertainty estimates using calibration of prediction intervals (see Figure 3 of [1]). A similar baseline would verify that the uncertainty learned by UMAL models is useful and that the improved test log-likelihoods given in Table 1 are not only the result of a more expressive model. Minor Comments: - What was the learning procedure for the mixture models in Table 1? The variance of the test log-likelihood for the synthetic dataset is surprisingly large. See, for example, the result for the three-component Laplace mixture. - How were asymmetry parameters selected at test time? The number of samples used during training is given in Line 212, but I could not find this information for the test procedure. Furthermore, the input to Algorithm 3 seems to imply that this variable is not marginalized using MC integration at test time, as it is during training. - Line 78 should read "... distribution, *a* fact that ..." - Line 169: "has" should be used instead of "have". - Line 175: "integrate" rather than "integrates". - Line 221: This sentence is hard to understand and should be re-phrased. What is the "50% random uniform generated data"? I understood this as 50% of the synthetic data was selected to form the training set, but that meaning is not obvious from the sentence. References: [1] Tagasovska, N., & Lopez-Paz, D. (2018). Frequentist uncertainty estimates for deep learning. arXiv preprint arXiv:1811.00908. [2] Dabney, W., Ostrovski, G., Silver, D., & Munos, R. (2018). Implicit quantile networks for distributional reinforcement learning. arXiv preprint arXiv:1806.06923. [3] Dua, D. and Graff, C. (2019). UCI Machine Learning Repository [http://archive.ics.uci.edu/ml]. Irvine, CA: University of California, School of Information and Computer Science.

Reviewer 2



This paper proposes a mixture of asymmetric laplacians (ALD) to model the distribution of an output variable for regression in deep learning. > The paper lacks clarity which is not helped by the dispersed typographic errors. The overuse of "uncountable mixtures" to refer to the simple process of marginalizing a 1D random variable, which is fed as input to a quantile regression network is confusing. > This work lacks novelty as the only contribution is quite simply a Monte Carlo average of a fixed number of ALDs. > The evaluation of the uncertainty measures crucially omits the concept of calibration and respective metrics. > This work is missing standard regression benchmarks such as the UCI datasets [1] or the benchmarks used in [2] (as well as prior work). I believe that the expressivity of this framework should allow for adequate performance on homogeneous distributions just as well as heterogeneous [1] Asuncion, Arthur, and David Newman. "UCI machine learning repository." (2007). [2] Lakshminarayanan, Balaji, Alexander Pritzel, and Charles Blundell. "Simple and scalable predictive uncertainty estimation using deep ensembles." Advances in Neural Information Processing Systems. 2017. ============ Having read the rebuttal, I appreciate the extensive experiments that empirically establish the calibration of the uncertainty measurement. Therefore, I updated my score but kept my comments as is for completeness.

Reviewer 3



Originality: The proposed UMAL is novel. It combines and extends the ideas of mixture density network and quantil regression, which treats each quantile level as a single component. It also draws connections to other quantil models. Quality: The technique used in this paper seems to be valid. The synthetic dataset experiment demonstrates the claimed advantages of UMAL. Clarity: When I first read the paper, I got stuck at the quantil regression section. So it would be great to include a brief review of the properties of quantil regression in the appendix. Also I am still a bit confused about the differences between Independent QR (IQR) model and Independent ALD, could you make it more clearer (such as the training objective function/training algorithm for IQR)? Significance: The proposed model empirically demonstrates the better modelling performance for the hetergeneous distribution. This might be a useful tool for modelling aleatoric uncertainties.

[Author Response · NeurIPS 2019]

First of all, we would like to thank all reviewers for their suggestions to improve our paper submission.

Reviewers **#1** and **#2** suggest experiments to measure if UMAL yields calibrated outputs. We have performed an additional empirical study to assess this point. We highlight that our system predicts an output distribution $p(y|x, \boldsymbol{w})$ (not a confidence value). In particular, we have computed the % of actual test data that falls into different thresholds of predicted probability. Ideally, given a certain threshold $\theta \in [0, 1]$, the amount of data points with a predicted probability above or equal to $1 - \theta$ should be similar to $\theta$. In the left part of Figure 1 we plot these measures for different methods (in green, our model) when considering the BCN RPF dataset. Furthermore, (following reviewers **#1** and **#2** suggestions) we have evaluated calibration quality on the UCI datasets with the same architectures as [2,3]. In the right part of Figure 1 we report the mean absolute error between the empirical measures and the ideal ones for all datasets. As it is shown, the conditional distribution predicted by UMAL has low error values. **Therefore, we can state that UMAL produces proper and calibrated conditional distributions that are especially suitable for heterogeneous problems**.

Figure 1: *Plot with the performance of three different models in terms of calibration. The mean and standard deviation for all folds of the mean absolute error between the predicted calibration and the perfect ideal calibration is represented in the table.*

| | BCN | YVC | Housing | Concrete | Energy | Kin8nm | Naval | Power | Protein | Wine | Yacht |
|---|---|---|---|---|---|---|---|---|---|---|---|
| N | .12±.04 | .04±.01 | .08±.04 | .03±.01 | .03±.01 | .02±.01 | .39±.02 | .02±.01 | .03±.00 | .03±.01 | .06±.02 |
| L | .03±.00 | .06±.01 | .07±.04 | .05±.02 | .04±.01 | .04±.01 | .35±.03 | .02±.00 | .06±.02 | .06±.02 | .07±.02 |
| I-QR | .10±.02 | .12±.02 | .20±.05 | .18±.02 | .15±.04 | .17±.01 | .12±.05 | .20±.02 | .06±.01 | .19±.03 | .14±.06 |
| 2N-MDN | .05±.02 | .12±.05 | .07±.05 | .04±.02 | .05±.02 | .01±.01 | .36±.04 | .03±.01 | .06±.01 | .08±.03 | .06±.02 |
| 3N-MDN | .07±.02 | .14±.04 | .07±.05 | .07±.03 | .04±.02 | .02±.01 | .37±.04 | .03±.01 | .11±.01 | .15±.04 | .07±.02 |
| 4N-MDN | .10±.03 | .17±.06 | .09±.05 | .10±.03 | .05±.02 | .03±.01 | .36±.05 | .03±.01 | .15±.01 | .18±.03 | .07±.04 |
| 10N-MDN | .19±.04 | .19±.06 | .12±.06 | .22±.06 | .09±.04 | .08±.01 | .33±.05 | .03±.01 | .22±.01 | .18±.02 | .09±.05 |
| 2L-MDN | .05±.01 | .09±.01 | .09±.05 | .06±.02 | .05±.02 | .04±.01 | .32±.04 | .07±.02 | .07±.00 | .06±.02 | .06±.03 |
| 3L-MDN | .08±.02 | .11±.02 | .11±.05 | .10±.03 | .08±.03 | .05±.01 | .29±.04 | .08±.02 | .12±.01 | .16±.03 | .06±.02 |
| 4L-MDN | .13±.05 | .12±.03 | .14±.06 | .12±.03 | .08±.04 | .06±.01 | .31±.04 | .07±.02 | .15±.01 | .15±.02 | .05±.02 |
| 10L-MDN | .24±.03 | .18±.05 | .21±.05 | .18±.04 | .16±.05 | .11±.01 | .27±.06 | .08±.02 | .22±.01 | .17±.01 | .10±.04 |
| I-$\mathcal{A}$LD | .06±.01 | .02±.01 | .07±.06 | .04±.01 | .05±.02 | .04±.01 | .44±.01 | .04±.01 | .07±.00 | .03±.01 | .09±.04 |
| UMAL | .04±.01 | .07±.01 | .10±.05 | .07±.04 | .06±.02 | .02±.01 | .43±.01 | .02±.01 | .02±.00 | .13±.06 | .06±.03 |

For the sake of completeness, we also have computed UMAL negative log-likelihood for UCI datasets (see Table 1) following [2]. These results restate that UMAL is always in the best positions. However, it should be noted that most of these databases have a **small sample size** and that in this regime, aleatoric uncertainty cannot be reliably estimated. We hypothesize that a better solution would be to simultaneous estimate epistemic (as [1,2,3]) and aleatoric uncertainty.

**Minor comments**:

- Quantile Regression allows us to approximate a desired quantile, unlike the classical regression that only estimates the mean or the median. This is useful since we could capture confidence intervals without strong assumptions about the distribution function to approximate (**Rev. #3. Q5.1**).

- IQR is a function, $\phi(\mathbf{x}, \tau)$, that can be evaluated for any real value of $\tau \in (0.1)$ to give us a point-wise estimation of an infinite number of quantiles $(\mu_\tau)_{\tau \in (0,1)}$ (**Rev. #3. Q5.3**). On the other hand, I-$\mathcal{A}$LD is a function, $\phi(\mathbf{x}, \tau)$, that predicts the $(\mu_\tau, b_\tau)_{\tau \in (0,1)}$ parameters of each ALD that is identified with a real asymmetry value, $\tau$. Thus, now each ALD tries to estimate, in a non-point-wise manner, their corresponding quantile (**Rev. #3.Q5.2**).

- If we consider each ALD as a component of a single mixture model we arrive to UMAL and, in turn, we solve the crossing quantiles problem because all ALDs are optimized jointly to produce the output distribution (**Rev. #3. Q5.4**).

- To obtain the distribution predicted by UMAL we will evaluate the learnt function $\psi(\mathbf{x}, \tau)$ for each $\tau$ in any discretitzation of its interval of definition, $(0, 1)$. For instance, by considering the partition $sel_\tau = [0.01, 0.02, \dots, 0.99]$ as parameter of the function defined in Algorithm 3 of the paper (**Rev. #1. Q2.2**).

We are seriously considering the suggestions made by reviewers **#1** and **#2** regarding typos and polishing. Thus, the paper is being revised by several native English speakers to improve its flow.

Table 1: *Comparison of the Negative Log-Likelihood of the test set over different train-test folds proposed in* [3].

| | Housing | Concrete | Energy | Kin8nm | Naval | Power | Protein | Wine | Yacht |
|---|---|---|---|---|---|---|---|---|---|
| Normal distribution | 2.76±.34 | 3.20±.16 | 2.13±.24 | −1.15±.03 | −3.67±.01 | 2.83±.03 | 2.84±.03 | 1.05±.14 | 1.86±.31 |
| Laplace distribution | 2.59±.20 | 3.21±.13 | 2.06±.20 | −1.08±.04 | −3.73±.04 | 2.87±.03 | 2.74±.01 | 1.00±.08 | 1.54±.37 |
| Independent QR | 10.96±2.4 | 10.19±.95 | 9.45±1.3 | 9.22±.66 | 5.14±.89 | 8.39±.45 | 8.14±.52 | 12.30±.91 | 10.32±2.9 |
| 2 comp. Normal MDN | 2.74±.30 | 3.25±.21 | 2.02±.30 | −1.15±.05 | −3.66±.02 | 2.85±.05 | 2.56±.03 | 1.33±.61 | 1.55±.32 |
| 3 comp. Normal MDN | 2.68±.28 | 3.64±.28 | 2.30±.43 | −1.15±.05 | −3.66±.01 | 2.85±.04 | 2.90±.15 | 0.69±1.0 | 1.54±.52 |
| 4 comp. Normal MDN | 2.87±.46 | 3.74±.28 | 2.46±.39 | −1.12±.04 | −3.66±.03 | 2.86±.05 | 3.32±.11 | 0.52±.90 | 1.43±.36 |
| 10 comp. Normal MDN | 3.10±.46 | 5.64±1.1 | 3.03±.71 | −0.99±.06 | −3.64±.03 | 2.86±.04 | 4.94±.75 | 0.75±.95 | 1.75±.49 |
| 2 comp. Laplace MDN | 2.61±.23 | 3.28±.14 | 2.06±.30 | −1.10±.04 | −3.70±.06 | 2.91±.05 | 2.50±.03 | 0.59±.63 | 1.37±.42 |
| 3 comp. Laplace MDN | 2.65±.25 | 3.45±.16 | 2.30±.21 | −1.09±.03 | −3.66±.06 | 2.65±.06 | 2.95±.04 | −0.81±.70 | 1.39±.35 |
| 4 comp. Laplace MDN | 2.76±.42 | 3.57±.14 | 2.31±.35 | −1.10±.05 | −3.68±.06 | 2.93±.04 | 2.79±.08 | −0.65±.96 | 1.45±.35 |
| 10 comp. Laplace MDN | 3.17±.46 | 3.95±.34 | 2.80±.49 | −0.98±.07 | −3.62±.10 | 2.96±.05 | 3.46±.12 | 0.52±.74 | 1.63±.34 |
| Independent $\mathcal{A}$LD | 2.79±.56 | 3.87±.12 | 2.28±.11 | −1.00±.05 | −2.82±.01 | 2.89±.02 | 2.68±.01 | 1.01±.07 | 1.78±.41 |
| UMAL model | 2.59±.26 | 3.74±.15 | 2.13±.14 | −1.09±.03 | −2.81±.01 | 2.85±.03 | 2.40±.01 | 0.14±.70 | 1.41±.38 |

[1] Gal, Y. and Ghahramani, Z. Dropout as a bayesian approximation: Representing model uncertainty in deep learning. ICML, 2016.

[2] Lakshminarayanan, Balaji, et al. Simple and scalable predictive uncertainty estimation using deep ensembles. NIPS, 2017.

[3] J. M. Hernández-Lobato, et al. Probabilistic backpropagation for scalable learning of Bayesian neural networks. ICML, 2015.


[Meta-Review · NeurIPS 2019]

The contribution is well-motivated, resulting in a simple method that is easy to implement, and providing a well-calibrated estimate of the uncertainty in the reported experiments. The novelty is rather moderate, but the developments are rigorous, pointing towards approximations where necessary. Please make sure to release code.